# Automatic Skill Generation
# for Knowledge Graph Question Answering

Maria Angela Pellegrino[1] ✉, Mario Santoro[1,2],
Vittorio Scarano[1], and Carmine Spagnuolo[1]

[1] Dipartimento di Informatica, Università degli Studi di Salerno, Italy
{mapellegrino,vitsca,cspagnuolo}@unisa.it
[2] m.santoro75@studenti.unisa.it

**Abstract.** Knowledge Graphs are a critical source for Question Answering, but their potential may be threatened due to the complexity of their query languages, such as SPARQL. On the opposite side, Virtual Assistants have witnessed an extraordinary interest as they enable users to pose questions in natural language. Many companies and researchers have combined Knowledge Graphs and Virtual Assistants, but no one has provided end-users with a generic methodology to generate extensions for automatically querying knowledge graphs. Thus, we propose a community shared software framework to create custom extensions to query knowledge graphs by virtual assistants, unlocking the potentialities of the Semantic Web technologies by bringing knowledge graphs in the "*pocket*" of everyone, accessible from smartphones or smart speakers.

**Keywords:** Question Answering · Knowledge Graphs · Virtual Assistant · Software Framework

## 1 Introduction and Motivation

Knowledge Graphs (KGs), i.e., graph-structured knowledge bases, are fast becoming a key instrument in disseminating and exploiting knowledge, but their potential might be threatened by the complexity of their query languages, such as SPARQL, too challenging for lay users [2,16].

Natural Language (NL) interfaces can mitigate these issues, enabling more intuitive data access and unlocking the potentialities of KGs to the majority of end-users [10] by losing in expressiveness while gaining in usability. NL interfaces may provide lay users with question-answering (QA) features where users can adopt their terminology and receive a concise answer. Researchers argue that multi-modal communication with virtual characters is a promising direction in accessing knowledge [4]. Thus, many companies and researchers have combined KGs and Virtual Assistants (VAs) [1,5,8,9,12], but no one has provided end-users with a generic methodology to automatically generate extensions to query KGs.

To fill this gap, we propose a community shared software framework (a.k.a. generator) that enables lay users to create ready-to-use custom extensions for performing question-answering over knowledge graphs (KGQA) for any cloud

provider. Our proposal may unlock the Semantic Web technologies potentialities by bringing KGs in the *"pocket"* of everyone, accessible from smartphones or smart speakers. It is the first attempt, to the best of our knowledge, to empower lay users in actively creating VA extensions by requiring little/no technical skills in query languages and VA extension development.

## 2   Related work

KGQA is a research field widely explored in the last years [6,7]and DBpedia gains a particular interest in being accessed by friendly user interfaces [13,14,17].

KGQA requires matching an input question to a subgraph. The simplest case requires matching a single KG triple, and it is also called simple QA [3]. In contrast to it, the task of complex QA requires matching more than one triple in the KG [15]. We present an approach general enough to deal with both simple and more complex queries. At the moment, we mainly cover patterns related to single triples enhanced by class refinement, numeric filters, and sorting options.

Focusing on KGQA in VA, it is natively offered in well-established personal assistants, like Google Assistant and Alexa, which provides users with content from generic KGs (Google Search and Microsoft Bing, respectively). Moreover, the Semantic Web community invested in increasing VA capabilities by providing QA over open KGs (e.g., Haase et al. [8] proposed an Alexa skill to query Wikidata by a generic approach) or in domain-specific applications (e.g., Krishnan et al. [11] explored the NASA System Engineering domain while Machidon et al. [12] and Anelli et al. [1] focus on the Cultural Heritage (CH) domain). While these approaches are demonstrated on custom but specific KGs, we propose a generic approach and we openly publish VA extensions for general-purpose KGs (e.g., DBpedia and Wikidata) and CH KGs (use cases are available on GitHub[4]).

## 3   Virtual Assistant Extensions Generator

The proposed generator automatically creates VA extensions performing KGQA by requiring little/no technical skills in programming and query languages. It provides users with the opportunity to customize and generate ready-to-use VA extensions. The implemented process is graphically represented in Fig. 1.

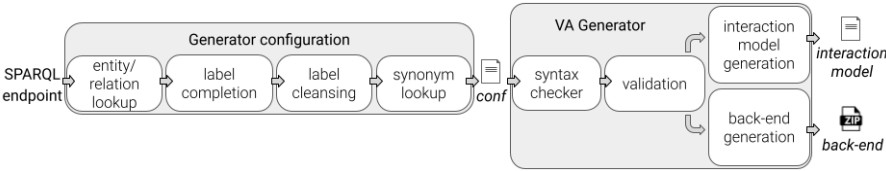

**Fig. 1.** Process implemented in the proposed generator to create a Virtual Assistant extension to perform question-answering over knowledge graphs.

The generator takes as input a configuration file that defines the SPARQL endpoint of interest, the VA extension language (`en` and `it` are supported at the moment), the invocation name (i.e., the skill wake-up word), and the list of desired intents. The implemented intents are tailored towards SPARQL constructs, and they cover `SELECT` and `ASK` queries, `class specification`, numeric `filters`, `order by` to get the superlative and path traversal. We model each supported SPARQL query template as an intent. Each intent is modelled by a set of utterances and can be completed by slot values. A custom slot represents each slot, and it requires the specification of a (complete or partial) set of values that it can assume. Therefore, users can specify entity and relation dictionaries to customize the set of values, and the mapping between entity and relation labels and their URIs. The generator checks the syntactical correctness of the configuration file during the `syntax checker` phase; validates the semantic correctness of the configuration during the `validation`; during the `interaction model generation`, it creates the `interaction_model.json` which contains configured intents, its utterances and the slot values as defined in the configuration file, while during the `back-end generation` phase, it produces the `back-end` (as a ZIP file) containing the back-end logic implementation. If any error occurs, the generator immediately stops and returns a message reporting the occurred error. If the configuration is properly provided, the generator returns a folder entitled as the skill wake-up word containing the `interaction model` as a JSON file and the `back-end` Node.js code as a ZIP file. The generated skill is ready to be used, i.e., it can automatically be uploaded on Amazon developer[3] and Amazon AWS[3], respectively. It corresponds to manually created skills, but our proposal may reduce required technical competencies and development time.

According to users' skills, they can provide the generator with a custom configuration file. Otherwise, they can exploit the `generator configuration` component that takes as input the SPARQL endpoint of interest, automatically retrieves both classes and relations labels and their URIs, and returns the configuration file that can be directly used to initialize the VA generator.

Each phase is kept separate by satisfying the modularity requirement, and it is implemented as an abstract module to enable extension opportunities easily. The actual version ($v1.0$), freely available on GitHub[4] is provided with a command-line interface and supports the Amazon Alexa provider.

As a future direction, we aim to enhance our generator by providing it with a (simple) graphical user interface and improve the intent identification, by using a Named Entity Recognition and Relation Extraction mechanism, and performing a disambiguation approach during the linking phase.

## 4   Virtual Assistant Extension Usage

In a VA-based process (see Fig. 2), users pose a question in NL by pronouncing or typing it via a VA app or dedicated device (e.g., Alexa app/device).

---

[3] Links for Alexa skill deployment: `developer.amazon.com` and `aws.amazon.com`
[4] `https://github.com/mariaangelapellegrino/virtual_assistant_generator`

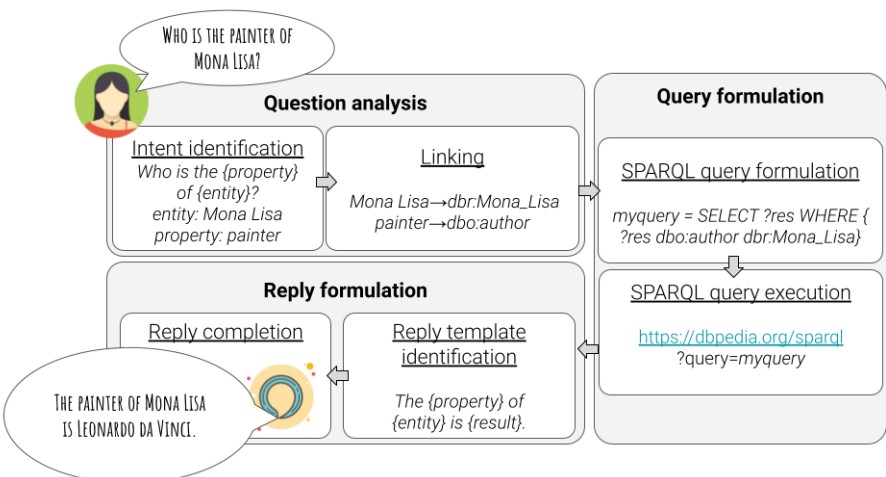

**Fig. 2.** Process to use VA extension for KGQA.

VAs are provided with an NL processing component to analyse questions. Thus, they perform intent identification to recognize the intent that matches the user query and solve the intent slots. Then, the back-end has to perform the (entity and relation) linking task, i.e., determine the URIs corresponding to the used labels. It can be accomplished by consulting a lookup dictionary or by calling an API service. Once completed the *question analysis* step, we can move to the *query formulation* phase. The back-end has to recognize the SPARQL pattern that fulfills the user request, formulate SPARQL query, and then run it over the SPARQL endpoint. Once results are returned, the VA extension can perform the *reply formulation* step, i.e., the reply template is identified and completed by actual results. Finally, the reply is returned to the user.

## 5   Demonstration

As a demonstration example[5], we show that users can automatically create an Alexa skill for querying well-known KGs, such as DBpedia. Thus, we see how to exploit the `generator configuration` and the `VA generator` components, we discuss the required steps to upload the skill on Alexa service providers, and we demonstrate the skill in practice by posing questions on Alexa Developer Console. The demo can reply to questions like *Who is the creator of goofy? How tall is Michael Jordan? Can you define Madama Butterfly?* to retrieve the object value of a KG triple; *How many programming languages are there?* as a special case of class refinement; *Which movie has producer equals to Hal*

---

[5] Demo link: `http://automatic_skill_generation_for_KGQA-DEMO-ESWC2021.mp4`

*Roach?* to retrieve the subject of KG triples; *Which library has established before 1400?* representing of a numeric filter; *Which is the river with maximum length?* modeling superlatives; *Can you check if Goofy has Walt Disney as creator?* representing an ask query.

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
