# OpenReview forum: "Automatic Skill Generation for Knowledge Graph Question Answering"
_eswc-conferences.org/ESWC/2021/Conference/Poster_and_Demo_Track — ESWC2021 P&D_

### Official Review · AnonReviewer4 · 2021-04-08
**A well presented and interesting paper which needs some minor polishment and clarifications**

**Rating:** 7
**Confidence:** 4

**Review:**

This paper presents a framework for automatic skill generation of question-answering over KGs. The presented demostration shows how giving a SPARQL endpoint the application is able to generate all the code needed to deploy an Amazon Alexa skill which gives effective answers collected from a targeted KG.

While the ideas of using Natural Language to build SPARQL queries and the use of Virtual Assistans as KG query interface are not new, I find the proposed framework really interesting as it lowers the barrier of required development background and the time consumed to create them. Indeed, I think that this framework supposes an interesting asset to KG builders as it offers another approach to query their data with a minimum effort on their side. However, I have some concers that I would like to discuss with the authors:

You use as an argument in favor of your system the complexity of query languages, like SPARQL. While, indeed I could agree on this affirmation, I really think that a cite can support better your arguments. In addition, the goals of the two interfaces are quite different. On one side, SPARQL aims to cover a full query language, with its inhereted flexibility and expressivity. On the other side, VA systems try to give a natural interface for humans, therefore they are not as expressive and flexible as query languages. So, maybe the problem is that the objective is not the same in both approaches.

This takes me to another related question. Which level of complexity are able to effectively translate your automatic generated skills? Can I pose questions that would require to cross different data or that would require nested queries?

You repeatedly say that the use of your framework requires "little/no-technical skills". However, looking to the video demonstration, you use the command line to automatically generate the skill. This seems totally the opposite of having no-technical skills. Same goes for the possibility to edit a JSON file. I would say that only a minimum set of technical skills are needed or that it supposes a minimum effort. Maybe you could enhance this aspect providing a simple user interface in further versions.

I was also wondering about supported languages because they are not mentioned in the paper. I found them in the Github repo but I think that is worth a mention inside the paper.

I missed a small related work section in which you compare your automatically generated outputs with other state-of-the-art solutions that you cite or other Natural Language to SPARQL converters. The idea is to see if NL capabilities integrated in Alexa can deliver better results than other approaches.

While it is not necessary, I would have really enjoyed having a skill deployed (e.g., Dbpedia one) in order to test directly in my Echo speaker. In addition, it would enhance a lot your demo and further presentations of your work.

Typos:

Abstract:

to querying -> to query OR for querying

**Anonymity:**

Yes, I would like my review to remain anonymous.

---

### Official Review · AnonReviewer3 · 2021-04-14
**Article can be presented as a visual demo**

**Rating:** 7
**Confidence:** 2

**Review:**

This paper seems to provide a conceptual approach to question answering utilizing knowledge graphs. This line of research can be considered as very up-to-date. The authors have provided a video showing the use of the system from the Alexa developer console.

The article misses discussion of existing related work on question answering with knowledge graphs, such as:

 Mynarz, Jindřich, and Václav Zeman. "DB-quiz: a DBpedia-backed knowledge game." Proceedings of the 12th International Conference on Semantic Systems. 2016.

Singh, Kuldeep, et al. "No one is perfect: Analysing the performance of question answering components over the DBpedia knowledge graph." Journal of Web Semantics 65 (2020): 100594.

Minor notes:
Before checking the external links to the video  and code (in a footnote), I had the impression that the development is in a less progressed stage than it actually is. In a revised version, the authors should focus on making the article more concrete. For example, I could not find information on for which knowledge graphs/use cases are templates readily available.

**Anonymity:**

Yes, I would like my review to remain anonymous.

---

### Official Review · AnonReviewer1 · 2021-04-15
**Review for Automatic Skill Generation for Knowledge Graph Question Answering**

**Rating:** 6
**Confidence:** 4

**Review:**

This paper proposes a software framework, i.e., generator, that allows users to create virtual assistant question answering extensions over knowledge graphs. Given a config file, the main steps towards creating the extension are 1) checking syntactical the correctness of the config file, 2) validating the semantic correctness of the config file, 3) interaction model generation 4) back-end generation. The created extension can then be used in, for instance, Amazon Alexa.

Having a framework that facilitates creating an extension for querying a knowledge base is indeed an interesting task and can be quite useful for many companies as well. The proposed framework takes as input a configuration file containing the SPARQL endpoint, the language of interest, intends of interest, and the mappings between entities' and relations' labels to their respective URIs should be provided. After validating the correctness of the config file, a back-end zip file is created. When a user poses a question in natural language, some steps are followed. First is the indent identification and linking. I believe that both of these components suffer from shortages. The indent identification part is nice but limits the users from more complex questions. Using a Named Entity Recognition and Relation Extraction seems a natural choice. For the linking part, it is proposed to use a dictionary that maps labels to their URIs. However, this can lead to considerable incorrect links and non-existence links in case the entity, or relation does not exactly correspond to the label (key of dict). A natural choice is to disambiguate the entities and relations. Next, the SPARQL queries are formulated and executed, and the result is returned.

In general, I am positive about the usefulness of the proposed framework. However, in the pipeline, much can be improved for better performance, as well as the novelty of the approach.


---- Some Minor remarks----
The paper's writing is nice, except for the abstract and introduction that have many sentences just repeated.





**Anonymity:**

Yes, I would like my review to remain anonymous.

---

### Official Review · AnonReviewer2 · 2021-04-15
**A tool to make knowledge graph question answering accessible to virtual assistants**

**Rating:** 8
**Confidence:** 4

**Review:**

Summary:
In this paper, the authors propose an automated generator that builds a bridge between question answering over KGs and virtual assistants. The authors intend to make queries over the KGs affordable by layman over smartphones or smart speakers. An end-to-end demo of the work has been provided for this work.

Pros:
- The paper is well written and the intermediate steps of the generator creation are well explained.
- The idea to bridge the gap between KGs and virtual assistants is novel via a generator.
- The tool also gives the flexibility to adapt to different KGs with different underlying structure.
- The demo is well documented.
- A screencast on the usage of the tool is also provided by the authors.

Cons:
- There is no mention and discussion on the differences with the state of the art models which involves DBpedia question answering over a UI, such as Clover Quiz [1]
- It would have been interesting to see an example from a KG from another domain, for e.g., the biomedical domain.

[1] Vega-Gorgojo G. Clover Quiz: a trivia game powered by DBpedia. Semantic Web. 2019 Jan 1;10(4):779-93.

Overall, I accept the paper because it presents an interesting demo in which the KG meets the virtual assistants and will be interesting to be discussed at the conference.


**Anonymity:**

Yes, I would like my review to remain anonymous.

---

### Decision · Program_Chairs · 2021-04-19

Accept